# The Penta-EF-Hand ALG-2 Protein Interacts with the Cytosolic Domain of the SOCE Regulator SARAF and Interferes with Ubiquitination

**DOI:** 10.3390/ijms21176315

**Published:** 2020-08-31

**Authors:** Wei Zhang, Ayaka Muramatsu, Rina Matsuo, Naoki Teranishi, Yui Kahara, Terunao Takahara, Hideki Shibata, Masatoshi Maki

**Affiliations:** Department of Applied Biosciences, Graduate School of Bioagricultural Sciences, Nagoya University, Furo-cho, Chikusa-ku, Nagoya 464-8601, Japan; qiangbantds@gmail.com (W.Z.); ayaka.cl613@gmail.com (A.M.); r.matsuo0115@gmail.com (R.M.); teranishi.naoki@gmail.com (N.T.); kahara.yui@c.mbox.nagoya-u.ac.jp (Y.K.); takahara@agr.nagoya-u.ac.jp (T.T.); shibabou@agr.nagoya-u.ac.jp (H.S.)

**Keywords:** adaptor, calcium-binding protein, HiBiT, NEDD4, *PDCD6*, protein-protein interaction, SOCE, ubiquitin, ubiquitination, WW domain

## Abstract

ALG-2 is a penta-EF-hand Ca^2+^-binding protein and interacts with a variety of proteins in mammalian cells. In order to find new ALG-2-binding partners, we searched a human protein database and retrieved sequences containing the previously identified ALG-2-binding motif type 2 (ABM-2). After selecting 12 high-scored sequences, we expressed partial or full-length GFP-fused proteins in HEK293 cells and performed a semi-quantitative in vitro binding assay. SARAF, a negative regulator of store-operated Ca^2+^ entry (SOCE), showed the strongest binding activity. Biochemical analysis of Strep-tagged and GFP-fused SARAF proteins revealed ubiquitination that proceeded during pulldown assays under certain buffer conditions. Overexpression of ALG-2 interfered with ubiquitination of wild-type SARAF but not ubiquitination of the F228S mutant that had impaired ALG-2-binding activity. The SARAF cytosolic domain (CytD) contains two PP*X*Y motifs targeted by the WW domains of NEDD4 family E3 ubiquitin ligases. The PP*X*Y motif proximal to the ABM-2 sequence was found to be more important for both in-cell ubiquitination and post-cell lysis ubiquitination. A ubiquitination-defective mutant of SARAF with Lys-to-Arg substitutions in the CytD showed a slower degradation rate by half-life analysis. ALG-2 promoted Ca^2+^-dependent CytD-to-CytD interactions of SARAF. The ALG-2 dimer may modulate the stability of SARAF by sterically blocking ubiquitination and by bridging SARAF molecules at the CytDs.

## 1. Introduction

Eukaryotes have a group of proteins that possess a domain of five serially repeated helix-loop-helix Ca^2+^-binding motifs (penta-EF-hand proteins, PEF proteins) [1]. Although the PEF protein ALG-2 (gene name: *PDCD6*) was originally identified as one of the factors associated with cell death of T cells [2], the roles of ALG-2 in apoptosis are not clearly understood. De-regulation of *PDCD6* gene expression has been reported in cancer cells [3,4,5,6] and serves as a potential prognostic marker for certain types of cancer [7]. ALG-2 was first shown to bind to an ALG-2-interacting protein named AIP1/ALIX [8,9], which was later shown to play roles as one of key associating factors of endosomal sorting complex required for transport (ESCRT) by binding to CHMP4s and TSG101 [10,11,12]. Further studies revealed that ALG-2 interacts Ca^2+^-dependently with a variety of cellular proteins involved in various processes including signal transductions, membrane trafficking, and membrane repair ([13,14,15], see Refs. [16,17] and references therein). Searching for new target proteins is necessary to gain a deeper insight into how ALG-2, which lacks catalytic domains, works in cells. We previously employed a bioinformatics approach to search for new ALG-2-interacting partners and successfully identified PATL1 and CHERP [18,19]. Our strategy was based on the fact that most, but not all, of the identified ALG-2-interacting proteins possess Pro-rich regions (PRRs) with some conserved sequences [20]. Our subsequent studies including sequence comparison and mutational analysis of ALG-2-binding sites based on X-ray crystal structures revealed the presence of three different ALG-2-binding motifs (ABMs) in the PRRs (See Ref. [17]) for review: type 1 (ABM-1, represented by ALIX); type 2 (ABM-2, represented by Sec31A); type 3 (ABM-3), MetPro repeats in IST1.

In the present study, we retrieved protein sequences containing the ABM-2 sequence from the UniProt database, selected high-scored sequences, expressed full-length or partial sequences as SGFP2 (strongly enhanced GFP-2)-fused proteins in human embryonic kidney 293 (HEK293) cells, and performed in vitro binding assays. Among the new candidates, the strongest binding activity was shown for SARAF, which was previously identified as an endoplasmic reticulum (ER)-transmembrane protein (TMEM66) that had a regulatory role in store-operated calcium entry (SOCE) [21]. SARAF was found to respond to cytosolic Ca^2+^ elevation after ER Ca^2+^ refilling by promoting a slow inactivation of SOCE activity by association with components of SOCE apparatus. Thus, SARAF is likely to protect cells from Ca^2+^ overfilling [21]. Since SARAF does not possess Ca^2+^-binding motifs, it has remained unknown how this negative regulator of SOCE senses cytosolic Ca^2+^ elevation. In this study, we demonstrated a Ca^2+^-dependent interaction between SARAF and ALG-2 by co-immunoprecipitation (co-IP) assays of the endogenous proteins in mammalian cells. We further characterized the interaction by overexpressing SGFP2-fused or Strep-tagged proteins of wild-type (WT) SARAF and various SARAF mutants. Unexpectedly, ubiquitination of SARAF was found to proceed not only inside cells but also during pulldown assays after cell lysis as evidenced by modification of SARAF with exogenously supplemented ubiquitin that was tagged with a nanoluciferase-split moiety named HiBiT [22]. ALG-2 interfered with ubiquitination of SARAF by binding to the ABM-2 sequence. Generally, a PP*X*Y motif serves as the binding site for WW domains of NEDD4 family E3 ligases that possess typical domain structures of the C2 domain, multiple WW domains and a catalytic domain homologous to the E6-AP carboxyl terminus (HECT) [23,24]. We investigated roles of the two PP*X*Y motifs present in the SARAF cytosolic domain (CytD) for ubiquitination and clarified the relationship between ALG-2 binding to SARAF and ubiquitination interference. A ubiquitination-impaired SARAF mutant with Lys-to-Arg substitutions in the CytD showed a decrease in the degradation rate in HEK293 cells by half-life analysis.

Recent X-ray crystal structural analysis of the SARAF luminal domain has revealed a domain-swapped β-sandwich fold that contributes to dimerization of SARAF [25]. We have found that the ALG-2 dimer bridges two SARAF molecules in a Ca^2+^-dependent manner by binding to the respective CytDs and we discussed the correlation with ubiquitination interference.

## 2. Results

### 2.1. Screening for Novel ALG-2-Interacting Proteins

In order to find new ALG-2-interacting partners, we first searched a protein sequence database of UniProt (20209 human entries) and retrieved 225 human protein sequences that contained ABM-2 {(P/Φ)P*X*(P/Φ)G(F/W)Ω} {(P/Φ), Pro or hydrophobic; (F/W), Phe or Trp; Ω, large side chain; *X*, variable; a higher content of Pro preferred} [26]. We selected the top 12 high-scored sequences for in vitro binding assays (See Appendix A for UniProt IDs, expressed regions, and summary of binding activities).

We expressed SGFP2-fused full-length proteins or partial fragments in HEK293 ALG-2 knockdown (ALG-2KD) cells and performed immunoprecipitation (IP) with a rabbit anti-GFP polyclonal antibody (pAb). IP of SGFP2-fused proteins was confirmed by Western blotting (WB) with a mouse anti-GFP monoclonal antibody (mAb) (Figure 1A, lower panel). As shown in the upper panel of Figure 1A, Far-western (FW) analysis using nanoluciferase (NanoLuc, Nluc)-fused ALG-2 as a probe revealed signals for Sec31A (positive control), DIAPH3, FNDC3A, cTAGE5, SHISA4 and SARAF but not for SGFP2 (negative control) and other proteins. A semi-quantitative pulldown assay was performed by incubating Nluc-ALG-2 with the expressed proteins that were immobilized to magnetic beads via an antibody and protein G, and the amounts of bound Nluc-ALG-2 were quantified by measuring luciferase activities (Figure 1B). Among the FW-positive SGFP2-fused proteins, SARAF-p showed the highest activity (71.5%). The values of binding activities to Sec31A-pΔ (0.58%) and DIAPH3-p (0.37%) were not statistically significant compared with the negative control (SGFP2, 0.18%). All of the proteins that showed strong or moderate positive interactions by the pulldown assay (Sec31A, SARAF, SHISA4, and cTAGE5; ≥4% relative binding) each have the ABM-2 sequence in a wide range (~50 a.a.) of predicted intrinsically disordered regions (Appendix A). ALG-2 may access the ABM-2 sequences that are already exposed in the disordered regions where unmasking processes are not required.

### 2.2. Co-Immunoprecipitation (co-IP) of Endogenous ALG-2 and SARAF

To investigate whether ALG-2 interacts with SARAF in human cells, we performed co-IP assays using respective specific antibodies. Since a commercially available anti-SARAF pAb did not work in our study, we prepared antisera raised in rabbits using the GST-fused SARAF CytD lacking a portion of the sequence including the ALG-2-binding motif (Figure 2A). 

The endogenous level of SARAF in HEK293 cells (*Input*) was too low to be unambiguously detectable by the affinity-purified anti-SARAF pAb by WB. However, WB after IP with anti-SARAF pAb in the presence of 100 μM CaCl_2_ (*Ca*) or 5 mM EGTA (*Eg*) produced clear bands corresponding to SARAF, the signals of which were greatly increased by overexpressing untagged SARAF by DNA transfection of pcDNA3-SARAF (Figure 2B, lower panel). Bands corresponding to SARAF were not observed by IP with control (*Ctrl*) IgG. WB with anti-ALG-2 pAb revealed faint (*vector*) or clear (SARAF overexpression) signals for the IP products obtained in the presence of CaCl_2_ but not in the presence of EGTA (Figure 2B, upper panel). As shown in Figure 2C, the amounts of co-immunoprecipitated ALG-2 with anti-SARAF pAb were greatly reduced to the control vector-transfected level when overexpressed SARAF mutants lacking the ABM-2 sequence PPPPGFK (ΔABM-2) or having amino acid substitution of phenylalanine with serine (F228S) were analyzed. Co-IP of ALG-2 with endogenous SARAF by anti-SARAF pAb in the presence of CaCl_2_ was observed without overexpression of untagged SARAF in HeLa cells and in breast cancer MCF7 cells (Figure 2D). WB signals of SARAF were stronger when lysates from MCF7 cells were used than when lysates from HeLa cells were used. 

To further confirm specific interaction between ALG-2 and SARAF, reciprocal co-IP was performed using lysates of HEK293 cells stably overexpressing untagged SARAF. As shown in Figure 3A, WB signals of SARAF were detected for the IP products with anti-ALG-2 mAb in the presence of 100 μM CaCl_2_ (*Ca*) but not in the presence of 5 mM EGTA (*Eg*). IP with control mouse IgG (*Ctrl*) gave no specific signals for ALG-2 or SARAF, confirming specific Ca^2+^-dependent interaction between ALG-2 and SARAF in the HEK293 cell lysates. To detect interaction between endogenous ALG-2 and SARAF proteins, anti-ALG-2 mAb was used for IP with lysates from HeLa cells and, for a negative control, from ALG-2 knockout (ALG-2KO) HeLa cells that had been established previously by the CRISPR-Cas9 system [27]. As shown in the top panel of Figure 3B, WB signals of anti-ALG-2 rabbit pAb were detected for the IP products with anti-ALG-2 mAb obtained from parental HeLa cells but not from ALG-2KO cells, indicating specificity of the antibodies. WB signals of SARAF were detected for the IP products from parental cells obtained in the presence of CaCl_2_ (*Ca*) but not in the presence of EGTA (*Eg*) (middle panel). ALG-2KO cells gave no WB signals of SARAF in the IP products obtained under both conditions. WB signals of Sec31A, analyzed as a positive control, showed similar results (bottom panel). Figure 3C shows the Ca^2+^-dependency of ALG-2 interaction with SARAF using MCF7 cell lysates. Addition of CaCl_2_ to the cell lysates was not necessary for the interaction, but inclusion of 5 mM EGTA caused loss of the interaction. This fact suggests that Ca^2+^ derived from the cells and/or a trace of Ca^2+^ contaminating the lysis buffer was sufficient for the interaction and that deprivation of Ca^2+^ leads to dissociation of the SARAF/ALG-2 complex formed inside cells.

### 2.3. Essential Region in SARAF for Interaction with ALG-2

Since the anti-SARAF pAb used in the present study was raised against the CytD, there still remained the possibility that the antibody might have interfered with binding of ALG-2 to additional unidentified sites other than the ABM-2 sequence, if any exist, in the CytD. As shown in Figure 4A, there are three regions rich in Pro and aromatic residues designated regions 1, 2 and 3. Region 2 corresponds to the ABM-2 sequence and regions 1 and 3 have ABM-1-like sequences. 

We expressed an epitope-tagged CytD (StrepHA-SARAF_CytD-SGFP2) and its various mutants in HEK293 cells and performed Strep-pulldown assays. Proteins bound to StrepTactin-immobilized magnetic beads (*Strep-pulldown*) were subjected to WB with anti-ALG-2 pAb (Figure 4B, upper panel) and with anti-HA mAb (lower panel). A strong WB signal of ALG-2 was detected in the pulldown products of WT StrepHA-SARAF_CytD-SGFP2. WB signals were below the detectable level in the region 2 mutant (Δ2, equivalent of ΔABM-2) and other deletion mutants, indicating that the ABM-2 sequence represents a single major ALG-2-binding site in SARAF.

### 2.4. Ubiquitination of SARAF

During the course of the experiments using various deletion mutants of SARAF (Figure 4B), we noticed that WB of the Strep-pulldown products with anti-HA gave faint multiple slower migrating bands in addition to the main bands in WT, Δ2, and Δ2,3 but not in Δ1,2 and Δ1,2,3 (Figure 4B, lower panel). Signals detected with anti-ubiquitin mAb gave similar ladder bands except for the major bands detectable with anti-HA (Appendix A). 

### 2.5. Suppression of Ubiquitination by Overexpression of ALG-2

In Figure 4B, the degree of ubiquitination in the Δ2 mutant appears to be more enhanced than that of the WT. On the other hand, additional deletion of the sequence (Δ1, 201-PPPYSEYPPF) caused loss of ubiquitination. To further clarify the relationship between ALG-2-binding ability and roles of region 1 for ubiquitination, we investigated the effects of amino acid substitutions of the CytD sequence on ubiquitination efficiency. Various types of amino acid-substituted mutants of StrepHA-SARAF_CytD-SGFP2 (Figure 5A) were each co-overexpressed with FLAG-tagged ALG-2 in HEK293 ALG-2KO cells and analyzed by a Strep-pulldown assay. As shown in Figure 5B, overexpression of FLAG-ALG-2 markedly suppressed the appearance of slower migrating bands (indication of ubiquitination) detected with anti-HA mAb. However, the overexpression had little effect on the F228S mutant, which had lost its binding ability to FLAG-ALG-2 as indicated by a loss of WB signals with anti-FLAG in the pulldown products. Unexpectedly, the ubiquitination and suppression by overexpressed ALG-2 were still observed in the mutant in which lysines in the CytD were all substituted with arginines (K229/326R, designated KR1/2). This fact indicates that major ubiquitination sites are present in regions other than the CytD in the expressed StrepHA-SARAF_CytD-SGFP2 protein. 

Since the CytD of SARAF contains two potential PP*X*Y motifs, which are known to serve as the binding site for WW domains of NEDD4 family E3 ligases [23], we investigated the effects of mutations at the two potential E3 ligase-associated PP*X*Y motifs (Figure 5A, 201-PPPY and 296-PPSY) by substituting Pro with Ala. As shown in Figure 5B, ubiquitination was significantly reduced for the PA1 mutant (P201/202A), reduced slightly for the PA2 mutant (P296/297A), and completely lost for the PA1/2 mutant (P201/202/296/297A). Thus, the sequence 201-PPPY located proximal to the transmembrane (TM) domain of SARAF is the major functional PP*X*Y motif of NEDD4-family E3 ligases. 

We also investigated which residues in the ALG-2 molecule are important for interaction with SARAF and for suppression of ubiquitination. FLAG-tagged ALG-2 WT and various mutants were co-expressed with StrepHA-SARAF_CytD-SGFP2 in HEK293 ALG-2KO cells, and Strep-pulldown products were analyzed as shown in the two upper panels of Figure 5C. WB signals for ALG-2 were not detected for mutants of E47/114A (Ca^2+^-binding defective mutant of EF1 and EF3; see Ref. [28]) and F85A (Phe-85, known to make contact with Sec31A ABM-2 residues; see Ref. [26]). These mutants did not interact with StrepHA-SARAF_CytD-SGFP2 or suppress its ubiquitination, suggesting the requirement of ALG-2 binding to the ABM-2 sequence for ubiquitination interference. Interestingly, the dimerization-impaired mutant Y180A [29], retaining SARAF-binding ability, showed reduced ubiquitination interference.

### 2.6. Evidence of In Vitro Ubiquitination of Expressed SARAF Constructs after Cell Lysis

The ratios of WB signal intensities of slower migrating bands to those of the main unmodified bands were greater in the Strep-pulldown products than in the input samples (Figure 5). We assumed that most ubiquitination reactions proceeded during manipulation of Strep-pulldown after cell lysis (overnight incubation at 4 °C). When lysates of cells expressing StrepHA-SARAF_CytD-SGFP2 were prepared using a buffer containing EDTA or were pre-treated at a high temperature (65 °C) caused loss of ubiquitination (Appendix A). 

In order to obtain evidence of post-cell lysis ubiquitination, we employed a NanoLuc luciferase complementation assay based on the high affinity interaction between the split NanoLuc luciferase large fragment (LgBiT) and the small peptide HiBiT that is fused with a protein of interest [22]. As shown in the schematic diagram of Figure 6, HiBiT-tagged ubiquitin (HiBiT-Ub) and StrepHA-SARAF_CytD-SGFP2 were expressed separately in HEK293 cells, and lysates (HKM buffer containing 0.2% Nonidet P-40) were combined before Strep-pulldown assays in the presence or absence of supplemental chemicals (EGTA, EDTA, CaCl_2_, NEM, or ATP plus DTT). Signals of HiBiT-Ub were remarkably reduced and completely lost in the presence of 5 mM EDTA and 10 mM NEM, respectively. Intensity of WB signals detected with anti-HA mAb correlated with HiBiT signals except for enhanced HiBiT-Ub signals in the presence of 3 mM ATP and 0.2 mM DTT for an unknown reason. WB signals for endogenous ALG-2 that should be co-pulled down with StrepHA-SARAF_CytD-SGFP2 were not detected under the experimental conditions of the pulldown buffer containing Ca^2+^-chelators (EGTA, EDTA). 

The in vitro ubiquitination marked with HiBiT-Ub during Strep-pulldown was also observed for StrepHA-tagged full-length SARAF (designated StrepHA-SARAF) in which StrepHA was inserted between the signal peptide (SP) sequence (1–30 a.a.) and the mature sequence (31–339 a.a.) of SARAF (Appendix A). Ubiquitination likely occurs at lysine residues not only in the CytD but also in the luminal region (luminal SARAF domain and linker region of StrepHA) that should be exposed to E3 ligases under non-physiological experimental conditions including solubilization of transmembrane proteins with a non-ionic detergent (1% Nonidet P-40). Consistent with the results of SGFP2-fused-SARAF CytD (Figure 5B), substitution of double prolines with alanines (P201/202A, designated PA1 mutant) in full-length SARAF (StrepHA-SARAF) remarkably reduced the signals of HiBiT-Ub (Appendix A).

### 2.7. In-Cell Ubiquitination of SARAF by NEDD4 Family E3 Ligases

A protein-protein interactome database obtained by high throughput affinity-capture mass spectrometry (MS) [30] suggests that SARAF interacts with WWP1 and ITCH (also named AIP4), members of WW-domain-containing NEDD4 family E3 ligases [23,31]. Although the degree of ubiquitination of SARAF is overestimated by WB of Strep-pulldown products due to the aforementioned post-cell lysis enzymatic reactions, WWP1 and ITCH are potential enzymes that physiologically catalyze ubiquitination of SARAF inside the cells. To investigate in-cell ubiquitination, we co-expressed StrepHA-SARAF with SGFP2-fused WWP1, WWP2 and ITCH (schematic diagrams presented in Figure 7A,B) and performed Strep-pulldown assays after lysing the cells with a buffer containing 5 mM EDTA and 10 mM NEM, which were demonstrated to block ubiquitination during the pulldown procedure (Figure 6 and Appendix A). 

As shown in Figure 7C, compared to co-expression with control SGFP2 protein (*Ctrl*), co-expression with WT SGFP2-WWP1 or SGFP2-ITCH enhanced anti-HA WB signals of multiple slower migrating bands corresponding to ubiquitinated StrepHA-SARAF in both samples of input and pulldown products. On the other hand, WWP1 C890A mutant in which the catalytic Cys-890 was substituted with Ala reduced signals to less than the control, probably by dominant-negative effects on endogenous E3 ligases. WWP2, tested in comparison with WWP1, showed no enhancing or suppressing effect. In the pulldown products, all SGFP2-fused E3 ligases were detected. To further analyze physical interactions between WWP1 and StrepHA-SARAF, we performed a Strep-pulldown assay by expressing catalysis-incompetent mutants (C890A and ΔHECT) and PP*X*Y-motif mutants of SARAF. As shown in Figure 7D, WB signals with anti-GFP were lost for PA1 (201-AAPY) and PA1/2 (201-AAPY, 296-AASY) and remarkably reduced for PA2 (296-AASY) compared with WT (201-PPPY, 296-PPSY). Overexpression of FLAG-ALG-2 was also effective on suppression of the WWP1-dependent in-cell ubiquitination (Appendix A). 

### 2.8. Slower Degradation Rate of Ubiquitination-Resistant SARAF Mutant

Ubiquitination of proteins is known to regulate various biological processes including proteasomal degradation, membrane receptor downregulation, multivesicular body (MVB) sorting, autophagy, signal transduction, and DNA repair [32,33,34]. We investigated whether ubiquitination of SARAF influences the half-life of the protein. To quantitatively measure the amount of SARAF, we used a HiBiT system. 

As schematically shown in Figure 8A, a HiBiT sequence followed by an HA-tag was inserted between the signal peptide sequence and the *N*-terminus of the mature protein of SARAF (31-339 a.a.). The C-terminus of SARAF was fused with a modified Strep tag {designated Strep(KR); Lys replaced with Arg}. Although detection by HiBiT blotting was less sensitive than WB with anti-HA mAb under our experimental conditions, SARAF mutants showed similar patterns of ubiquitinated bands in the samples of total cleared cell lysates (input) and pulldown products (Appendix A). No ubiquitination was observed in the mutant KR1/2 in which both of the Lys residues in the CytD were replaced with Arg, indicating absence of non-physiological ubiquitination in the luminal domain. For half-life analysis, the HiBiT-tagged SARAF proteins of the WT and KR1/2 mutant were each co-expressed with SGFP2-WWP1 in HEK293 ALG-2KO cells. The amounts of HiBiT-tagged proteins remaining after treatment with cycloheximide (CHX, protein synthesis inhibitor) in the presence or absence of thapsigargin (TG) were quantified by the HiBiT lytic assay system as shown in Figure 8B. The degradation rate for the KR1/2 mutant was slightly slower than that for the WT (half-life, ~2 h in the absence of TG) as observed at 2 h after CHX addition. The difference in the effect of TG on the degradation rate was small and not statistically significant, but there was a tendency of faster degradation for the WT.

### 2.9. Enhanced Ca^2+^-Dependent CytD-to-CytD Interaction by ALG-2

ALG-2 forms a dimer and bridges two interacting proteins as a Ca^2+^-dependent adaptor protein [35]. To investigate whether the CytD of SARAF may interact each other, we co-expressed HiBiTHA-SARAF and StrepHA-SARAF_CytD-SGFP2 of WT or F228S (ALG-2 binding-defective mutant) in HEK293 cells and performed co-IP with anti-GFP pAb followed by the HiBiT assay with LgBiT (Figure 9A). The amounts of immunoprecipitated GFP-fused proteins were normalized by the StrepTactin-alkaline phosphatase (AP) assay. As shown in Figure 9B, HiBiTHA-SARAF WT was co-immunoprecipitated with StrepHA-SARAF_CytD-SGFP2 WT, representing CytD-to-CytD interaction, but not with StrepHA-SGFP2 (control). The amount of co-immunoprecipitated HiBiTHA-SARAF F228S, was significantly reduced (~15% of WT; WT expressed as 100%) under the condition of a lysis buffer containing 10 μM Ca^2+^ (*Ca*). Since StrepHA-SGFP2 showed very low interaction activities (<0.3%), the SARAF CytD may interact with the same domain of the counterpart molecule directly or indirectly through other proteins in the SOCE apparatus in the absence of Ca^2+^. Enhancement of the Ca^2+^-dependent interaction between the WT SARAF CytDs compared to the Ca^2+^-independent interaction (in the presence of 5 mM EGTA; *Eg*) was significantly reduced in HEK293 ALG-2KO cells (fold enhancement: parental cells, 3.7 fold; ALG-2KO cells, 1.3 fold; data not depicted). As shown in Figure 9C, the Ca^2+^-dependent interaction was recovered by co-overexpressing FLAG-ALG-2 WT (*Ca*, 100%; *Eg*, 12.3%) but not by co-transfection of the expression vector for the FLAG-ALG-2 F85A mutant (impaired in interaction with SARAF, see Figure 5C) or empty FLAG vector (*Ca*, 12.6%; *Eg*, 12.3%). The recovery rate by co-overexpression of FLAG-ALG-2 Y180A (dimerization-impaired mutant of ALG-2, see Ref. [36]) was small (*Ca*, 24.1%; *Eg*, 7.7%). 

## 3. Discussion

ALG-2 interacts with a variety of proteins in a Ca^2+^-dependent manner and plays modulatory roles in diverse physiological phenomena [16]. Since ALG-2 contains no catalytic domains, physiological functions have been suggested by identification of interacting proteins. With combination of a bioinformatics approach and in vitro binding assays, we identified SARAF, an ER-transmembrane protein associated with SOCE, as a novel ALG-2 binding partner. Unexpectedly, we observed substantial amounts of ubiquitinated StrepHA-SARAF_CytD-SGFP2 proteins in the Strep-pulldown products (Figure 4 and Figure 5). The observed ubiquitination occurred mostly after cell lysis during Strep-pulldown as revealed by inhibition of ubiquitination by E3 ligase inhibitors (EDTA and NEM) and by modification with exogenously supplemented HiBiT ubiquitin (Figure 6). Interestingly, ubiquitination was dependent on the PP*X*Y motif proximal to the transmembrane (TM) domain and was suppressed by overexpression of ALG-2 (Figure 5). Although remarkable ubiquitin modifications of SARAF were observed in vitro during Strep-pulldown after cell lysis, the results of WB analysis using the total cell lysate or using the products of Strep-pulldown performed in the presence of E3 ligase inhibitors (EDTA and NEM) suggested that ubiquitination of SARAF proceeded within cells to a certain degree (Figure 7). Among the three NEDD4 family E3 ligases tested in the overexpression assay, WWP1 and ITCH but not WWP2 accelerated ubiquitination. These results may be due to the fact that amino acid sequences of WWP1 and ITCH are more similar each other than those of WWP1 and WWP2 [31]. In-cell ubiquitination of full-length SARAF was also dependent on PP*X*Y motifs (Appendix A). Co-overexpression with FLAG-ALG-2 WT caused reduction in the amounts of SARAF that was ubiquitinated both inside cells and by post-cell lysis (Figure 5C and Appendix A), but co-overexpression with SARAF-binding impaired mutants of FLAG-ALG-2 had no effects (Figure 5C, E47/114A and F85A). Interestingly, WWP1 contains ABM-2-like sequences in its WW domains (382-PLPPGWE, 457-PLPPGWE). However, inhibition of SARAF ubiquitination by direct binding of ALG-2 to these sites in the WW domains is unlikely because ubiquitination of an ALG-2 binding-defective mutant of SARAF (F228S) was not suppressed by ALG-2 (Figure 5B and Appendix A). Instead, ALG-2 may suppress ubiquitination by NEDD4 family E3 ligases through sterically blocking binding of the WW domains of E3 ligases to the PP*X*Y motif proximal to the TM domain. The degree of ubiquitination inhibition by the ALG-2 Y180A mutant was reduced despite the fact that it retained SARAF-binding activity (Figure 5C). Since Tyr-180 is a critical residue for dimerization of ALG-2 [29] and for bridging interacting partners [36], the difference in bulkiness of ALG-2 (monomer *vs* dimer) may account for the efficiency of inhibition. The distance between the functional PP*X*Y motifs and the ALG-2-binding sites may account for the efficiency of ubiquitination inhibition by ALG-2 (SARAF: PPPY to ABM-2, ~20 a.a.). Interestingly, one of the two lysine residues (potential ubiquitination sites) in the SARAF CytD is present in the ABM-2 sequence (Lys-229). It remains to be established whether masking of this residue by ALG-2 also contributes to interference of ubiquitination in cells. 

The estimated degradation rate of SARAF WT was greater than that of the ubiquitination-impaired KR1/2 mutant in the presence of TG (relative amount of HiBiT-tagged SARAF remaining 2 h after CHX addition: WT, 42.7%; KR1/2, 61.2%) (Figure 8). However, the estimated degradation rate after initial 2 h period was decreased and the differences became obscure. In this study, we used the HiBiT technology to quantitate the amount of remaining HiBiT-fused SARAF. The lytic assay that was based on complementation of LgBiT with HiBiT may not have distinguished intact and partially degraded HiBiT-fused proteins. It remains to be clarified whether the tendency for a greater degradation rate of SARAF in the presence of TG is associated with the SOCE and/or activation of NEDD4 family E3 ligases. Influx of extracellular Ca^2+^ through the SOCE system is achieved by the interplay of ER-transmembrane STIM1 and plasma membrane CRAC channel representatively comprised of Orai1 [37,38]. The SOCE apparatus is regulated by a number of modulators that physically associate with STIM1 or Orai1 [39]. Cytosolic EF-hand Ca^2+^ sensor proteins such as calmodulin [40,41,42,43], CRACR2A [44], and EFHB [45] are involved in the activation or feed-back inhibition of CRAC channels. SARAF, an ER-transmembrane protein, is a negative modulator of SOCE and physically associates with the SOCE apparatus [21,46,47] or TRPC1 [48]. Since SARAF does not contain any Ca^2+^-binding motifs, regulation of SARAF itself should depend on other Ca^2+^-dependent mechanisms. In this regard, we assumed that ALG-2 functions as a modulator of intracellular Ca^2+^ homeostasis through action on SARAF. Although we attempted to investigate effects of overexpression of ALG-2 or those of overexpression of ALG-2 binding-impaired mutant of SARAF F228S on calcium dynamics in HEK293 cells by a calcium imaging technique, we have not been able to observe significant effects. More reliable Ca^2+^ monitoring analysis under better experimental conditions are needed to clarify the effects of ALG-2 on Ca^2+^ dynamics in relation to SARAF function in the SOCE system. Fluorescence microscopic analysis of Ca^2+^-dependent spatio-temporal association of ALG-2 with SARAF and the SOCE apparatus in cells may be necessary to understand physiological functions of ALG-2.

Kimberlin et al. [25] reported that SARAF dimerizes at the luminal domain with a new domain-swapped β-sandwich fold. The effect of Ca^2+^ on dimerization of SARAF were not investigated by those authors. We performed a co-IP assay by using expression constructs of full-length SARAF fused with different tags (SGFP2 and HiBiT). There were no significant differences between WT and F228S and between immunoprecipitation conditions in the presence and absence of Ca^2+^ (data not shown), supporting the preformed dimerization at the luminal domain. However, by using an expression construct of the SARAF CytD and full-length SARAF, we observed an increase in the amount of co-immunoprecipitated full-length SARAF by IP of StrepHA-SARAF_CytD-SGFP2 with anti-GFP pAb in the presence of Ca^2+^ (Figure 9). The F228S mutation at the ALG-2-binding site in the SARAF CytD did not increase the amounts of co-immunoprecipitated full-length SARAF regardless of the presence of Ca^2+^ (Figure 9B). Furthermore, by similar experiments using ALG-2KO HEK293 cells, a Ca^2+^-dependent increase in the amounts of co-immunoprecipitated full-length SARAF was not observed (FLAG empty vector transfection), but co-overexpression of FLAG-ALG-2 WT caused recovery of the Ca^2+^-dependent increase in the CytD-to-CytD interaction (Figure 9C). Phe-85, which is located in Pocket 3 of ALG-2 and is essential for interaction with Sec31A [26] as well as with SARAF (Figure 5C), was also essential for the Ca^2+^-dependent CytD-to-CytD interaction because the F85A mutant did not recover the interaction in the complementation assay (Figure 9C). On the other hand, Tyr-180, which is located in Pocket 1 of ALG-2 and is capable of interacting with the ALIX peptide [29], was not important for interaction with SARAF as shown by the SARAF-binding activity of the FLAG-ALG-2 Y180A mutant (Figure 5C). Thus, the Ca^2+^/ALG-2-dependent CytD-to-CytD interaction of SARAF shown in Figure 9 was not due to a conformational change of the SARAF CytD induced by ALG-2 binding. However, FLAG-ALG-2 Y180A had little effect on the CytD-to-CytD interaction. Y180 is essential for the formation of an EF5-EF5 pairing interface of the ALG-2 dimer and formation of the bottom surface of Pocket 1 [29]. Thus, a dimer form of ALG-2 most likely contributes to bridging the CytDs of SARAF as a Ca^2+^-dependent adaptor in a manner similar to that previously shown for other interacting partners [35]. The clear difference in the inability of the Y180A mutant to function as an adaptor is that the mutant does not bridge ALIX, TSG101 and annexin A11 to the partner proteins due to its lack of binding ability in addition to its lack of dimerization potency, whereas this mutant retains binding ability in the case of SARAF and Sec31A. 

In conclusion, we identified SARAF as a novel interacting partner of ALG-2 and determined the ALG-2-binding site. Ubiquitination of SARAF by the NEDD4 family E3 ligase WWP1 was suppressed by co-overexpressing ALG-2, probably due to steric hindrance between ALG-2 and WWP1, which were found to have respective binding sites in the CytD at a close distance (~20 residues apart). A dimer of SARAF may be pre-formed at the luminal domain but stabilized by the ALG-2 dimer that binds at the CytD in the presence of Ca^2+^. 

## 4. Materials and Methods

### 4.1. Antibodies and Reagents

Anti-HA rat monoclonal antibody (mAb) (3F10, Roche Diagnostics, Basel, Switzerland), anti-GFP mouse mAb (B2) (Santa Cruz Biotechnology, Santa Cruz, CA, USA, sc-9996), anti-multi-ubiquitin mouse mAb (FK2) (MBL, D058-3), and anti-ALG-2 mouse mAb (2B4, Abnova, Taipei, Taiwan, H00010016-M01) were purchased. Rabbit polyclonal antibody (pAb) against ALG-2 was prepared previously [49]. Antisera against GFP and SARAF were obtained by immunizing rabbits with respective glutathione-*S*-transferase (GST)-fused proteins at Japan Lamb (Fukuyama, Hiroshima, Japan), and their polyclonal antibodies (pAbs) were affinity-purified using maltose-binding protein (MBP)-fused proteins immobilized to Sepharose columns as described in detail in Appendix A. Horseradish peroxidase (HRP)-conjugated goat antibodies against mouse or rabbit IgG and an HRP-conjugated rabbit pAb against goat IgG were obtained from Jackson ImmunoResearch (West Grove, PA, USA). Carbachol (sc-202092) was purchased from Santa Cruz Biotechnology (Dallas, TX, USA). Coelenterazine h was obtained from FUJIFILM Wako Pure Chemicals (Osaka, Japan). 

### 4.2. In Silico Screening

A database search for ABM-2-containing sequences was performed by the custom-made program “XaaRR-Scan” using a UniProt protein sequence database (http://www.uniprot.org/) as described previously except for excluding Pro content threshold at the initial screen [18]. From 20209 entries, we selected 12 human sequences of interest by scoring whether they contain 3 or more Pro residues in the ABM-2 sequence, presence of Pro at position 4, and presence of a sequence similar to that in the mouse proteins. ABM-2 sequences located in the predicted extracellular and luminal domains were excluded. 

### 4.3. Plasmid Construction

Human cDNAs used in the present study for novel ALG-2-interacting protein screening were obtained from GE Healthcare/Open Biosystems, NITE Biological Resource Center (NBRC) (Kisarazu, Chiba, Japan) and RIKEN Bioresource Center (Tsukuba, Ibaraki, Japan). SGFP2-fused expression plasmids were constructed by subcloning the cDNAs into the EcoRI/SalI site of pSGFP2-C1 [50] or into the EcoRI/BamHI site of pSGG-SGFP2-N1 by PCR cloning with an In-Fusion^®^ HD Cloning Kit (Clontech/Takarabio, Kusatsu, Shiga, Japan) or with a simple and ultra-low cost homemade seamless ligation cloning extract (SLiCE) [51] using specific primers (See Appendix A for database accession numbers, clone numbers, expressed regions and cloning sites in the vectors used.). Mammalian expression vectors of pStrepHA and pStrepHA-SGPF2 were constructed from pEXPR-IBA105 (Twin-Strep vector, IBA GmbH, Göttingen, Germany) by inserting a 3xHA tag at the C-terminal side of the Twin-Strep tag after modifications of intermediate constructs as described in Appendix A.

An expression vector for the SARAF CytD (195-339 a.a.) that was tagged with StrepHA and SGFP2 at the *N*-terminus and C-terminus, respectively, was constructed by the In-Fusion PCR cloning method (primers: forward, 5′-atctgagctcaagcttagtgacgggcagtat-3′; 5′-gcctccgcttggatctcgtctcctggtacc-3′), and the amplified DNA fragment was inserted between the HindIII and BamHI sites of pStrepHA-SGPF2. The resultant plasmid was designated pStrepHA-SARAF_ CytD-SGFP2. An expression vector for the full-length SARAF that was tagged with StrepHA located immediately downstream of the signal peptide sequence (designated pSP-StrepHA-SARAF; expressed protein name, StrepHA-SARAF) was constructed by insertion of a synthetic DNA block encoding the SARAF signal peptide (SARAF 1-30 a.a.) between the XbaI and NheI sites of pStrepHA-SGFP2 followed by replacement of the SGFP2-encoding DNA segment with the DNA fragment encoding SARAF 31-339 a.a. by the In-Fusion PCR method (See Appendix A for details.). A cytoplasmic domain-truncated SARAF-expression vector, pSP-StrepHA-SARAFΔCytD, was similarly constructed by replacement with the DNA fragment encoding SARAF 31-197 a.a. Site-directed mutagenesis to introduce a deletion of the ALG-2-binding site (ΔABM-2) and F228S substitution in SARAF was performed according to the instruction manual provided with a PCR-based QuikChange^®^ Site-Directed Mutagenesis Kit (Agilent Technologies Japan, Hachioji, Japan) using a combination of specific primers. 

An expression plasmid of three-tandemly repeated HiBiT-tagged ubiquitins (HiBiT-Ub) was constructed by fusing synthetic DNAs encoding the HiBiT amino acid sequence (VSGWRLFKKIS), GGS linker and ubiquitin and was inserted into the BamHI/EcoRI sites of pcDNA3.1 by a gene synthesis service at GenScript (Piscataway, NJ, USA). A nucleotide sequence for HiBiT-Ub has been registered at DNA databanks DDBJ/GenBank/EMBL (accession No. LC437023). An expression vector for HiBiT-HA-SARAF was constructed by multiple-step manipulation including insertion of a SARAF cDNA fragment into the pcDNA3.1-derived vector as described in Appendix A. 

### 4.4. Cell Culture and DNA Transfection

HEK293 cells were cultured in DMEM (Nissui) supplemented with 4 mM glutamine, 5% (or 10%) fetal bovine serum (FBS), 100 units/mL penicillin and 100 μg/mL streptomycin at 37 °C under humidified air containing 5% CO_2_. ALG-2 knockdown (KD) and knockout (KO) HEK293 cells were described previously [36,52]. Cells were seeded and cultured for one day, and then they were transfected with the expression plasmid DNAs by the conventional calcium phosphate method or by using the DNA transfection kit of FuGENE 6 (Roche) or PEI-MAX (Polysciences Inc. 24765, Warrington, PA, USA). Cultured cells were harvested and washed in PBS (137 mM NaCl, 2.7 mM KCl, 8 mM Na_2_HPO_4_ and 1.5 mM KH_2_PO_4_, pH 7.4) for biochemical analysis.

### 4.5. Preparation of Nluc-ALG-2

A synthetic gene encoding Nanoluciferase-fused ALG-2 (Nluc-ALG-2, DDBJ/GenBank/EMBL accession No. LC381846) for bacterial expression was synthesized and subcloned into the NcoI/EcoRI site of a T7 RNA polymerase expression vector, pET24d, at GenScript (Piscataway, NJ, USA). *Escherichia coli* Rosetta (DE3) was transformed with the recombinant plasmid and cultured at 20 °C overnight after induction for expression with 0.5 mM isopropyl β-D-1-thiogalactopyranoside (IPTG). Nluc-ALG-2 was purified using an ALG-2-binding peptide affinity column essentially as described previously [20,53]. 

### 4.6. Binding Assays with Nluc-ALG-2

HEK293 ALG-2KD cells were transfected with each expression plasmid encoding the respective SGFP2-fused protein. After 24 h, harvested cells were lysed at 4 °C with lysis buffer HKM (20 mM HEPES-NaOH, 142.5 mM KCl, 1.5 mM MgCl_2_) containing 0.2% Nonidet P-40, protease inhibitors (0.2 mM phenylmethylsulfonyl fluoride, 3 μg/mL leupeptin, 1 μM E64, 1 μM pepstatin, 0.1 mM pefabloc), and 10 μM EGTA for 30 min and then hand sonicated three times for 10 s each time (UR-20P, TOMY SEIKO, Tokyo, Japan). The cleared cell lysates obtained by centrifugation at 10,000× *g* for 10 min were used for immunoprecipitation by incubation with rabbit anti-GFP pAb overnight followed by incubation with Protein G magnetic beads (Dynabeads^®^ protein G, Thermo Fisher Scientific, Waltham, MA, USA) for 1 h. Collected magnetic beads were washed with buffer HKM containing protease inhibitors and EGTA. Aliquots of the suspensions of beads were subjected to Far-Western (FW) blotting and pulldown-binding assays essentially as described previously [53]. The membrane was incubated with 1 mL of coelenterazine h solution (20 μg/mL) in TBS containing 100 μM CaCl_2_, and the luminescence signal was detected with a LAS-3000mini lumino-image analyzer (FUJIFILM, Tokyo, Japan).

For pulldown-binding assay, aliquots of the suspensions of beads were incubated with 0.2 mL of NlucALG-2 solution (300 ng/mL) in lysis buffer HKM containing 0.2% Nonidet P-40, 0.1% gelatin, protease inhibitors, and 100 μM CaCl_2_ for 2 h at 4 °C. After the beads had been collected with a magnetic stand and washed three times with 0.5 mL of lysis buffer HKM containing 0.2% Nonidet P-40 and 100 μM CaCl_2_, the beads were suspended with 50 μL of passive lysis buffer (PLB) (Promega, Madison, WI, USA). Aliquots were taken and luciferase activities were measured using a Nano-Glo^®^ Luciferase assay reagent kit (Promega) with a luminometer (AB-2250, ATTO, Tokyo, Japan) or a photon-counting type microplate reader (Monochromator Multimode Microplate Reader-Berthold Technologies Mithras^2^ LB 943, Bad Wildbad, Germany). For normalization of the amounts of immunoprecipitated StrepHA-SGFP2-fused proteins, activities of StrepTactin-conjugated alkaline phosphatase (Precision Protein StrepTactin-AP Conjugate, BIO-RAD, Hercules, CA, USA) were measured using a substrate contained in a Phospha-Light™ SEAP Reporter Gene Assay System (Thermo Fisher Scientific, Waltham, MA, USA) as described previously [27,53].

### 4.7. Co-Immunoprecipitation (Co-IP) Assays

Cells were lysed by sonication in lysis buffer HKM containing 1% CHAPS and protease inhibitors, and the cleared lysates were incubated with respective antibodies in the presence of 100 μM CaCl_2_ or 5 mM EGTA overnight at 4 °C followed by incubation with Protein G magnetic beads (Dynabeads^®^ protein G). Proteins bound to the beads were analyzed by WB with specific antibodies.

### 4.8. Strep-Pulldown Assays

One day after HEK293 cells had been transfected with expression vectors for Strep-and 3xHA (StrepHA)-tagged SARAF proteins, the cells were harvested with PBS and lysed with lysis buffer HKM containing protease inhibitors and 0.2% Nonidet P-40 for the SARAF CytD or 1% Nonidet P-40 for full-length SARAF. The 10,000× *g* supernatants (cleared cell lysates) were incubated with Strep-Tactin^®^ XT- coated magnetic beads (MagStrep “type3” XT beads, IBA, 2-4090-002) at 4 °C overnight. Then the beads were collected using a magnetic stand and washed with the lysis buffer three times. The pulldown products were subjected to Western blotting by probing with anti-HA rat mAb and anti-ALG-2 rabbit pAb. When detection of in-cell ubiquitination of SARAF was required, A Strep-pulldown assay of full-length SARAF was performed under the condition of blocking post-cell lysis ubiquitination reactions by using the lysis buffer HK (20 mM HEPES-NaOH, 142.5 mM KCl) containing 2 mM EDTA, 10 mM *N*-Ethylmaleimide (NEM), 1% Nonidet P-40 and 20 μM MG132 in addition to other protease inhibitors.

### 4.9. Detection of HiBiT-Tagged Proteins

The amounts of HiBiT-tagged proteins in total cell lysates or in pulldown products were estimated with a Nano-Glo^®^ HiBiT Lytic Detection System (Promega N3030) that contained solutions of LgBiT protein, substrate and buffer. For detection of HiBiT-tagged proteins after SDS-PAGE, proteins were blotted onto a nitrocellulose membrane and subjected to the LgBiT reaction essentially according to the technical manual for a Nano-Glo^®^ HiBiT Blotting System (Promega N2410). Signals were detected with a LAS-3000mini lumino-image analyzer. 

### 4.10. Half-Life Analysis

After HEK293 ALG-2KO cells in a 9-cm culture dish had been transfected with an expression plasmid for WT or the ubiquitination site mutant (KR1/2) of HiBiT-SARAF-Strep(KR) and cultured overnight, the cells were re-suspended, re-plated into 24-well dishes, and cultured for overnight. CHX and TG were added to the culture media at the final concentrations of 20 μg/mL and 100 nM, respectively, and the culture was continued for the indicated period of time (0, 2, 4, and 6 h). The amounts of remaining HiBiT-fused SARAF proteins under each experimental condition in triplicated wells were assayed by the HiBiT lytic detection system as described above. Cells in 24-well plates were suspended and harvested with 75 μL of passive lysis buffer (PBL, provided by Promega) and stored overnight at −80 °C. LgBiT reaction was performed according to the instruction manual provided by the manufacturer, and chemiluminescent signals were measured with the multi-mode microplate reader Mithras^2^ LB 943 using a 384-well plate. 

### 4.11. Statistical Analysis

Statistical analysis was performed by one-way analysis of variance (ANOVA) followed by Tukey’s test using Origin 9.1 (Micro Software, Northampton, MA, USA). *p*-Values less than 0.05 are considered statistically significant.

### 4.12. Research Ethics

We followed biosafety guidelines for recombinant DNA research at Nagoya University. Experimental proposals were approved by the Recombinant DNA Biosafety Committee of the Graduate School of Bioagricultural Sciences, Nagoya University: Nou15-067 (approved on 24 March 2016) and Nou19-002 (approved on 12 April 2019). 

## Figures and Tables

**Figure 1 ijms-21-06315-f001:**
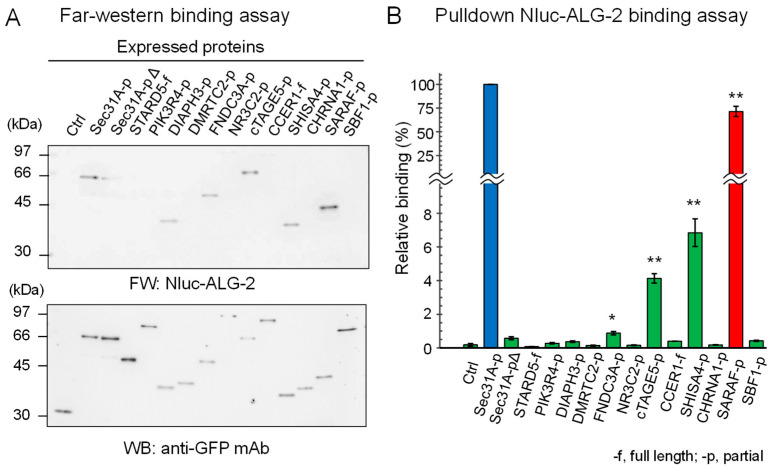
ALG-2-binding assays of SGFP2-fused proteins containing ABM-2 sequences. Full lengths (-f) or partial fragments (-p) of ALG-2-binding candidate proteins that were fused with SGFP2 or unfused negative control SGFP2 (Ctrl) were transiently expressed in HEK293 ALG-2KD cells. After SGFP2-fused proteins had been immunoprecipitated with anti-GFP pAb from cleared cell lysates as described in Materials and Methods, proteins bound to the protein G-immobilized magnetic beads were subjected to ALG-2-binding assays by (**A**) Far Western blotting (FW) and by (**B**) pulldown assays. (**A**) Blotted membranes were probed with Nluc-ALG-2 for FW (upper panel) and anti-GFP mAb (lower panel) for Western blotting (WB). (**B**) The amounts of Nluc-ALG-2 bound to the beads were quantified by measuring luciferase activities. Binding activities relative to Sec31A-p (100%) were calculated and data were expressed as mean ± SE (*n* = 3). Binding capacities compared to the negative control were statistically evaluated: * *p* < 0.01; ** *p* < 0.001.

**Figure 2 ijms-21-06315-f002:**
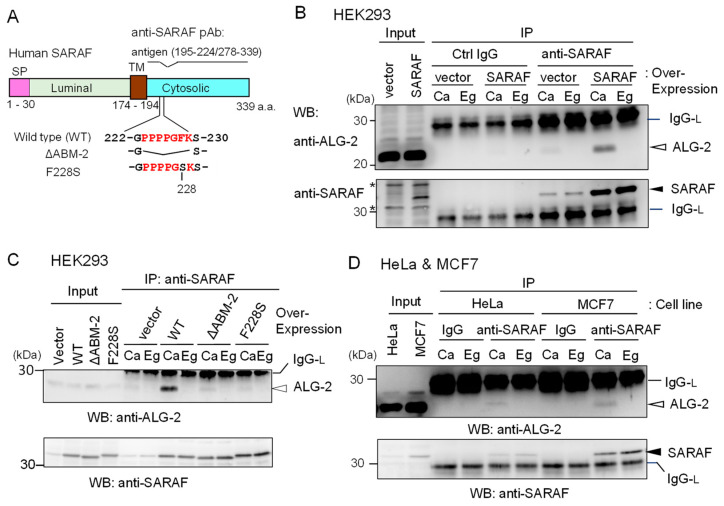
Co-IP of ALG-2 and SARAF with anti-SARAF polyclonal antibody. (**A**) Schematic diagram of SARAF showing the region of antigen used for polyclonal antibody (pAb) preparation and mutants lacking the sequence of the ALG-2-binding motif type 2 (ΔABM-2) or amino acid substitution (F228S). SP, signal peptide; TM, transmembrane. (**B**,**C**) Co-IP assays using HEK293 cells expressing untagged wild type (WT) SARAF or mutants (ΔABM-2, F228S) were performed as described in Materials and Methods. Antibodies (control rabbit *IgG*, anti-SARAF pAb) were added to the cleared cell lysates (*Input*) that were supplemented with 100 μM CaCl_2_ (*Ca*) or 5 mM EGTA (*Eg*), and immunoprecipitated proteins (*IP*) were analyzed by WB with anti-ALG-2 pAb (upper panel) and anti-SARAF pAb (lower panel). *IgG-L*, IgG light chain. Filled arrowhead, SARAF; unfilled arrow head, ALG-2; asterisks, non-specific. (**D**) HeLa cells and MCF7 cells were used for co-IP assays of endogenously present ALG-2 and SARAF as described in B and C except for no DNA transfection. *IgG*, control antibody.

**Figure 3 ijms-21-06315-f003:**
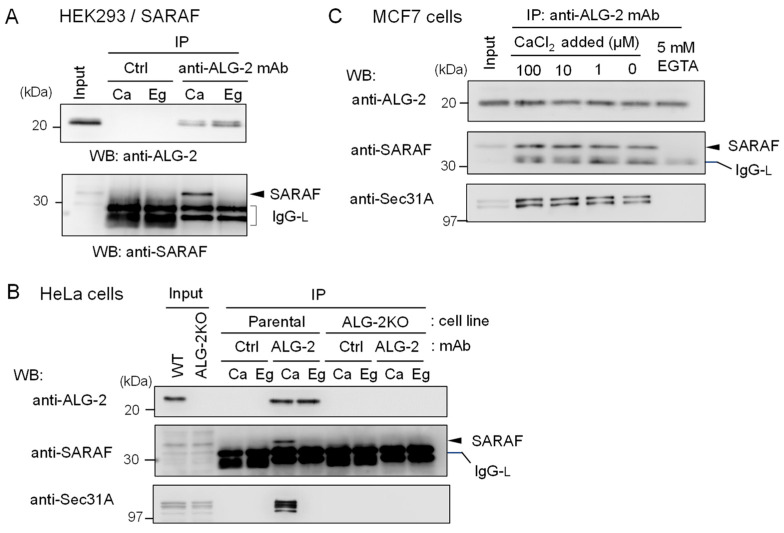
Reciprocal co-IP assay of ALG-2 and SARAF with anti-ALG-2 antibody. (**A**) HEK293 cells stably expressing untagged SARAF were used for co-IP assays in the presence of 100 μM CaCl_2_ (*Ca*) or 5 mM EGTA (*Eg*) with anti-ALG-2 mAb and control mouse IgG. Immunoprecipitates (*IP*) were analyzed by WB with anti-ALG-2 pAb and anti-SARAF pAb as indicated. *IgG-L*, IgG light chain. (**B**) Lysates of parental HeLa cells and ALG-2 knockout (KO) HeLa cells (ALG-2KO) were subjected to immunoprecipitation with anti-ALG-2 mouse mAb followed by WB with rabbit pAbs against ALG-2, SARAF and Sec31A as indicated. (**C**) A co-IP assay with anti-ALG-2 mAb was performed as shown in (**B**) but with the use of MCF7 cells in the presence of varying concentrations of exogenously added CaCl_2_ or 5 mM EGTA as indicated.

**Figure 4 ijms-21-06315-f004:**
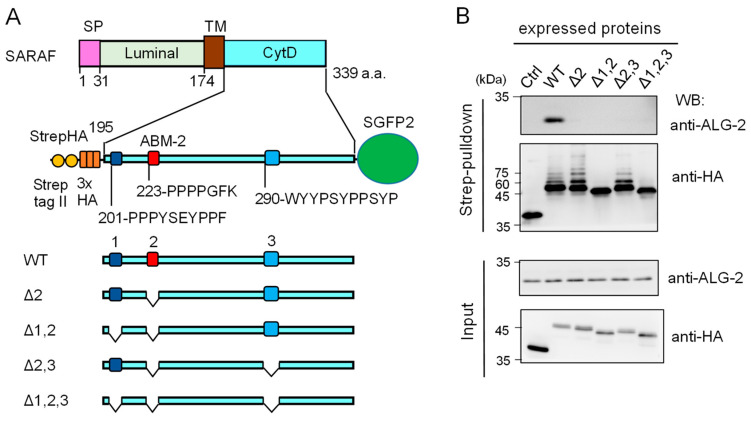
Deletion mutation of the SARAF cytosolic domain (CytD) affecting ALG-2 binding and ubiquitination. (**A**) Schematic diagram of the SARAF CytD, which contains three distinct motifs rich in Pro and aromatic residues (designated regions 1, 2 and 3). Expression plasmids for the SARAF CytDs of WT and various deletion mutants that were fused with Strep-tag II and 3xHA (StrepHA) at the *N*-terminus and SGFP2 at the C-terminus were constructed. (**B**) Strep-pulldown assay. After HEK293 cells had been transfected with pStrepHA-SARAF_CytD-SGFP2 (WT and deletion mutants) and cultured for 24 h, the cells were lysed with lysis buffer HKM containing protease inhibitors, 0.2% Nonidet P-40 and 10 μM CaCl_2_. The cleared cell lysates (*Input*) were subjected to Strep-pulldown followed by WB with anti-ALG-2 and anti-HA as indicated.

**Figure 5 ijms-21-06315-f005:**
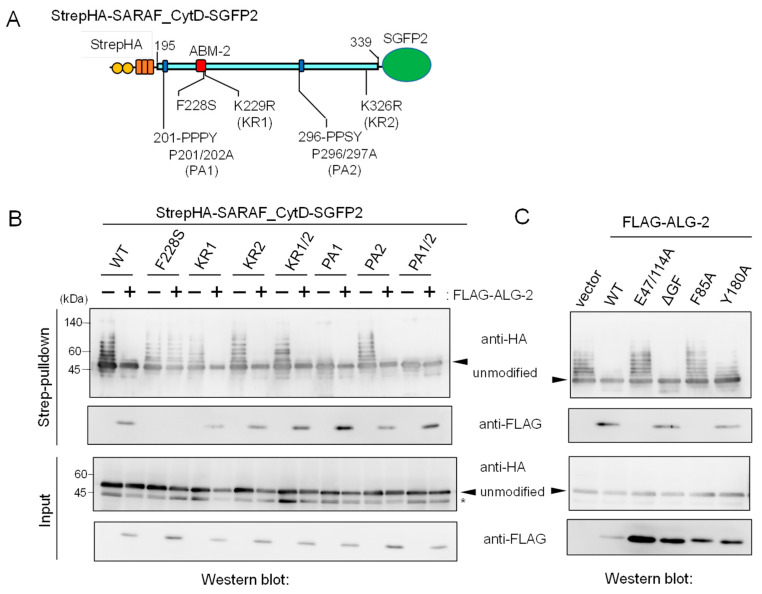
Mutations in the SARAF CytD affecting ubiquitination and suppression of ubiquitination by overexpression of ALG-2. (**A**) Schematic diagram of mutations at ABM-2 (F228S), two PP*X*Y motifs (Pro-to-Ala mutations) and two potential ubiquitination sites (Lys-to-Arg mutations) in the SARAF CytD. (**B**,**C**) After HEK293 ALG-2KO cells had been co-transfected with expression plasmids for StrepHA-SARAF_CytD-SGFP (WT or mutants) and with (**B**) FLAG-ALG-2 (WT) or with (**C**) FLAG-ALG-2 mutants and cultured for 24 h, the cleared cell lysates were prepared as described in the legend to Figure 4B and were subjected to the Strep-pulldown assay followed by WB with anti-HA and anti-FLAG antibodies as indicated. Asterisk, degraded products lacking a Strep tag or non-specific bands; arrowheads, non-ubiquitinated unmodified proteins.

**Figure 6 ijms-21-06315-f006:**
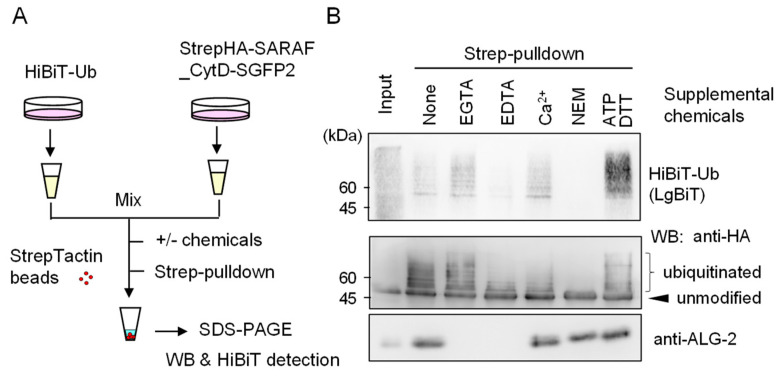
Evidence of ubiquitination reactions after cell lysis. (**A**) Schematic diagram of HiBiT-Ub assay. (**B**) StrepHA-SARAF_CytD-SGFP2 and HiBiT-tagged ubiquitin (HiBiT-Ub) were individually expressed in HEK293 cells in separate culture dishes. The cleared cell lysates prepared with HKM buffer containing 0.2% Nonidet-P were mixed, and aliquots were subjected to Strep-pulldown in the presence of supplemental chemicals as indicated (5 mM EGTA, 5 mM EDTA, 10 μM CaCl_2_, 10 mM NEM or 3 mM ATP plus 0.2 mM DTT). Pulldown products were resolved by SDS-PAGE followed by Western blotting with respective antibodies as indicated (anti-HA, anti-ALG-2) or by probing with LgBiT for detection of HiBiT-Ub.

**Figure 7 ijms-21-06315-f007:**
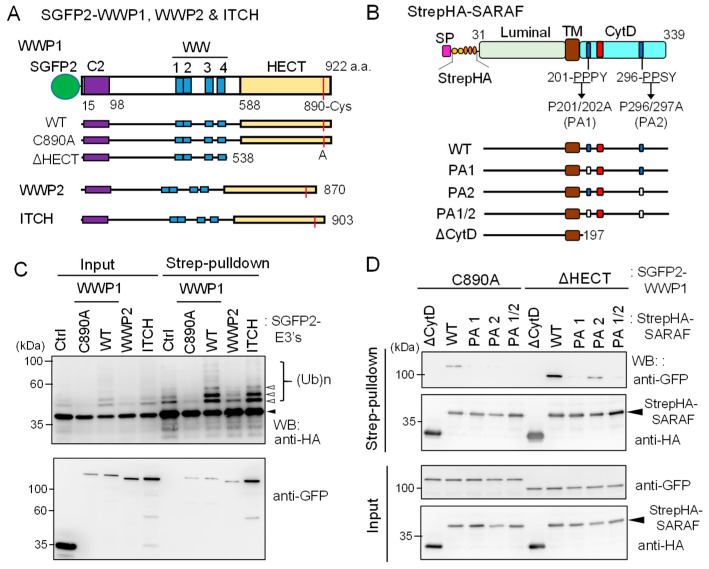
Ubiquitination of SARAF by WWP1 ubiquitin E3 ligase. (**A**) Schematic diagrams of SGFP2-fused NEDD4 family ubiquitin E3 ligases investigated in this study. The E3 ligases contain the Ca^2+^-binding C2 domain, four repeats of WW domains, and the catalytic HECT domain. The catalytic Cys residues in each E3 ligases are shown in red lines. The active site Cys of WWP1 was substituted with Ala (C890A) and the HECT domain was deleted (ΔHECT). (**B**) Schematic diagrams representing StrepHA-SARAF mutants of Pro-to-Ala substitutions at PP*X*Y motifs and the CytD truncation. (**C**) StrepHA-SARAF was co-expressed in HEK293 cells with SGFP2-fused E3 ligase WWP1 (WT or C890A mutant), WWP2, ITCH or with a control vector (*Ctrl*). The cells were lysed in lysis buffer HK containing 1% Nonidet P-40 and E3 ligase inhibitors (2 mM EDTA, 10 mM NEM) as well as the protease inhibitor cocktail supplemented with 20 μM MG132. The cleared cell lysate (*Input*) was subjected to Strep-pulldown followed by WB with anti-HA and anti-GFP mAbs. Unmodified and ubiquitinated StrepHA-SARAF bands are marked with an arrowhead and unfilled arrowheads, respectively. (**D**) Strep-pulldown assays were performed using StrepHA-SARAF mutants and E3 ligase-defective mutants of SGFP2-WWP1 (C890A or ΔHECT) to determine important regions in SARAF for interaction with WWP1.

**Figure 8 ijms-21-06315-f008:**
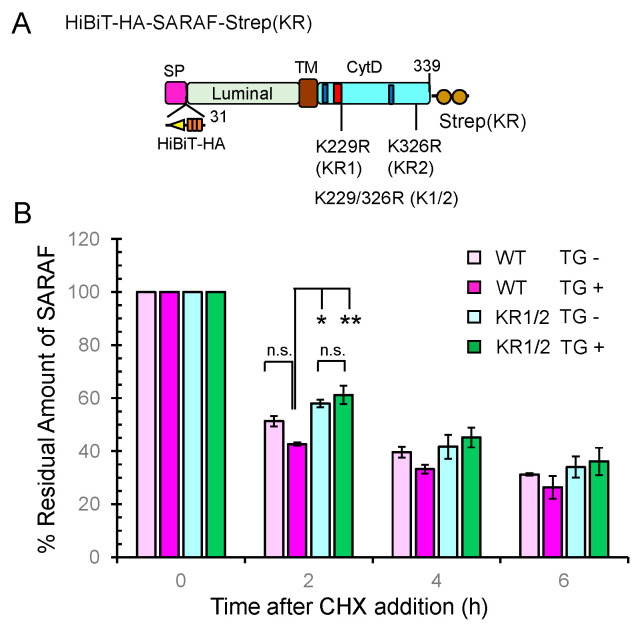
Slowdown of SARAF protein degradation rate by Lys-to-Arg substitutions in the CytD. (**A**) Schematic diagram of full-length SARAF that was tagged with HiBiT and 3xHA (designated HiBiTHA) between the signal peptide (SP) and the *N*-terminus of mature SARAF and tagged at the C-terminus with the Lys-to-Arg-substituted twin Strep (designated Strep(KR). (**B**) The half-life assays were repeated three times as described in Materials and Methods and the results are expressed as 100% at the 0 time for each condition. Data are expressed as mean ± SEM (*n* = 3). Statistical significance by Tukey’s test is indicated by asterisks (* *p* < 0.05; ** *p* < 0.01). n.s., not significant. CHX, cycloheximide; TG, thapsigargin.

**Figure 9 ijms-21-06315-f009:**
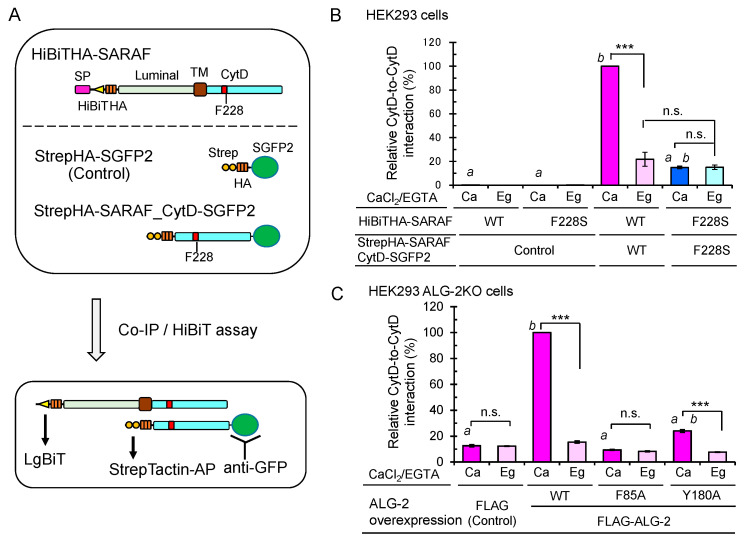
Importance of ALG-2-binding to SARAF for Ca^2+^-dependent interaction between the SARAF CytDs. (**A**) Schematic diagram of expressed proteins and co-IP/HiBiT assay. (**B**) After HEK293 cells in 6-cm dishes had been co-transfected with expression plasmids for full-length HiBiTHA-SARAF (WT or F228S mutant) and for either StrepHA-SGFP2 (Control) or StrepHA-SARAF_CytD-SGFP2 (WT or F228S mutant) and cultured for 24 h, the cells were lysed with lysis buffer HKM containing 1% Nonidet P-40 supplemented with protease inhibitors, 10 mM NEM and 10 μM CaCl_2_ (*Ca*) or 5 mM EGTA (*Eg*). SGFP2-fused proteins were immunoprecipitated with anti-GFP pAb, and the amounts of co-immunoprecipitated HiBiTHA-SARAF were estimated by the lytic HiBiT assay with LgBiT. The amounts of immunoprecipitated SGFP2-fused proteins were normalized by the StrepTactin-AP method as described in Materials and Methods. The relative amount of HiBiTHA-SARAF WT co-immunoprecipitated with StrepHA-SARAF_CytD-SGFP2 WT was expressed as 100% of relative interaction activity. The co-IP/HiBiT assay was performed in duplicate and repeated three times. Data are expressed as mean ± SEM (*n* = 3). Statistical significance by Tukey’s test is indicated in three different ways: asterisks for comparison between conditions of plus and minus Ca^2+^; *a* for comparison with WT in the presence of Ca^2+^; *b* for comparison with the control in the presence of Ca^2+^. *p* values were below 0.001 for all cases indicated (***, *a*, and *b*). n.s., not significant. (**C**) Enhancement of the Ca^2+^-dependent CytD-to-CytD interaction of SARAF in HEK293 ALG-2KO cells by co-overexpression of FLAG-ALG-2. HEK293 ALG-2KO cells were co-transfected with expression plasmids for (i) full-length HiBiTHA-SARAF (WT), (ii) StrepHA-SARAF_CytD-SGFP2 (WT) and (iii) FLAG empty vector or FLAG-ALG-2 (WT, F85A or Y180A) and cultured for 24 h. The cells were lysed and subjected to HiBiT assay as described above. Relative interaction by co-overexpression of ALG-2 WT in the presence of Ca^2+^ was expressed as 100% activity. Data are expressed as mean ± SEM (*n* = 3). Statistical significance by Tukey’s test is indicated by asterisks, *a*, and *b* as described in (**B**) for comparison with the presence or absence of Ca^2+^ (asterisks), WT (*a*) and control (*b*).

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
