# Peer review of "The Penta-EF-Hand ALG-2 Protein Interacts with the Cytosolic Domain of the SOCE Regulator SARAF and Interferes with Ubiquitination"

_ijms, 2020, doi:10.3390/ijms21176315_

Round 1

Reviewer 1 Report

The paper by Maki and collaborators describes the discovery of several novel ALG-2 binding proteins based on the known ALG-2 interaction motifs of established ALG-2 binders using bioinformatics. Several of these proteins were initially tested for binding to ALG-2. The one with the highest binding capacity tested in pulldown and far Western analysis, SARAF, was further biochemically analysed and the ALG-2 binding site was determined. Interestingly, ALG-2 binding to SARAF interfered with ubiquitination of the protein and might by this inhibit its degradation. Further, ubiquitination of SARAF was investigated and the sites of ubiquitination mapped and it was shown that the ubiquitination-resistant SARAF was degraded slower than the wild-type SARAF.

The paper is well done with all the necessary experiments and the controls. The conclusions are supported by the presented data. Most of the data is of biochemical nature and little information is presented on the biological significance of ALG-2 interaction with SARAF with the exception that ALG-2 interaction with SARAF might contribute to the stability of SARAF in a cellular environment. However, this is not shown in a cellular system with endogenous expression of SARAF in cells with or in the absence of ALG-2. Most important would of course be to find out whether ALG-2 would modulate the known activity of SARAF as a regulator of SOCE.

Major point: The article is difficult to read as the reader is distracted by extensive text that is not directly related and/or relevant for the message. It should therefore be considerably shortened to make it more readable even though the number of words may get below the minimal of 4000 what the journal requires (which is a non-scientific rule!). Results contain too many technical details and part of the discussion reporting on results presented only in Supplementary or "not shown" should be eliminated. Figure legends could be shortened: eg. buffer components listed in fig 1 and others should not be describes as this is part of the methods.

Minor point: The paper contains 24 self-citations and much of the contents in these papers has been reviewed by the group. Again, in order to make the paper more compact and easier to read, discussions and citations of work by the group that is not directly related to the article or has been reviewed should be eliminated.

In conclusion: This is a solid biochemical work and an important starting point to investigate the physiological significance of the ALG-2-SARAF interaction.

Author Response

Reviewer 1

Comments and Suggestions for Authors
The paper by Maki and collaborators describes the discovery of several novel ALG-2 binding proteins based on the known ALG-2 interaction motifs of established ALG-2 binders using bioinformatics. Several of these proteins were initially tested for binding to ALG-2. The one with the highest binding capacity tested in pulldown and far Western analysis, SARAF, was further biochemically analysed and the ALG-2 binding site was determined. Interestingly, ALG-2 binding to SARAF interfered with ubiquitination of the protein and might by this inhibit its degradation. Further, ubiquitination of SARAF was investigated and the sites of ubiquitination mapped and it was shown that the ubiquitination-resistant SARAF was degraded slower than the wild-type SARAF.

The paper is well done with all the necessary experiments and the controls. The conclusions are supported by the presented data. Most of the data is of biochemical nature and little information is presented on the biological significance of ALG-2 interaction with SARAF with the exception that ALG-2 interaction with SARAF might contribute to the stability of SARAF in a cellular environment. However, this is not shown in a cellular system with endogenous expression of SARAF in cells with or in the absence of ALG-2. Most important would of course be to find out whether ALG-2 would modulate the known activity of SARAF as a regulator of SOCE.

Response:  We thank Reviewer 1 very much for reading our manuscript and for giving us valuable comments to improve it. To our regret, we have not been able to clarify roles of ALG-2 in association with SARAF for calcium homeostasis modulation or for other biological functions in cells. However, we do hope that finding of interference with ubiquitination of SARAF by ALG-2 will contribute to further understanding of functions of the ALG-2 protein in the cell in the future.   

Major point: The article is difficult to read as the reader is distracted by extensive text that is not directly related and/or relevant for the message. It should therefore be considerably shortened to make it more readable even though the number of words may get below the minimal of 4000 what the journal requires (which is a non-scientific rule!). Results contain too many technical details and part of the discussion reporting on results presented only in Supplementary or "not shown" should be eliminated. Figure legends could be shortened: eg. buffer components listed in fig 1 and others should not be describes as this is part of the methods.

Response:  We agree that the article needs easy to read. According to Reviewer 1’s kind advice, we have shorten the manuscript by deleting description of too much technical details and discussions on supplementary figures only indicated in Discussion including calcium homeostasis. Legends of a few figures were also rewritten. Several figures presented as Supplementary Figures were deleted. In addition, a few Figures were rearranged and transferred from Main body to Supplementary material. Changes were made as follows:
Table 1, transferred to new Suppl Table S1. 
Fig 4C, transferred to new Suppl Fig S2
Fig 6A, transferred to new Suppl Fig S3
Fig 7, transferred to new Suppl Fig S4
Fig 8, renumbered to new Fig 7
Fig 9, renumbered to new Fig 8
Fig 10, renumbered to new Fig 9
Suppl Fig S1, renumbered to new Suppl Fig S6
Suppl Fig S2, renumbered to Suppl Fig S1
Suppl Fig S6, renumbered to Suppl Fig S5
Suppl Figs S3, S4, S5, S7, S8 and S9 were deleted.

Minor point: The paper contains 24 self-citations and much of the contents in these papers has been reviewed by the group. Again, in order to make the paper more compact and easier to read, discussions and citations of work by the group that is not directly related to the article or has been reviewed should be eliminated.

Response:  We thank Reviewer 1 again for the kind advice to improve our manuscript. We deleted self-citations as much as possible and replaced them with review articles. Citations of works by our group that are not directly related to the present article were deleted. Total numbers of references cited in the article were reduced from 73 to 53.

In conclusion: This is a solid biochemical work and an important starting point to investigate the physiological significance of the ALG-2-SARAF interaction.

Response:  We appreciate very much for the encouraging comments.

Reviewer 2 Report

The paper is well written. The authors have carried out a very thorough study and data presented is convincing. The major concern is that all studies are carried out with over-experession of the proteins. This does not confirm that any interactions demonstrated will occur with the levels of proteins that exist naturally in cells. I would have liked to have some data presented that demonstrates interaction at the cellular level proteins (e.g. using FRET etc.) This would be my major concern with the current study.  Adding such data will strengthen the impact of the study.

Author Response

Reviewer 2

Comments and Suggestions for Authors

The paper is well written. The authors have carried out a very thorough study and data presented is convincing. The major concern is that all studies are carried out with over-experession of the proteins. This does not confirm that any interactions demonstrated will occur with the levels of proteins that exist naturally in cells. I would have liked to have some data presented that demonstrates interaction at the cellular level proteins (e.g. using FRET etc.) This would be my major concern with the current study.  Adding such data will strengthen the impact of the study.

Response: We thank Reviewer 2 for reviewing our manuscript and giving us valuable suggestions. We disagree with the Reviewer’s statement that “all studies are carried out with over-experession of the proteins.” This statement is not correct. Co-immunoprecipitation assays with specific antibodies against SARAF and ALG-2 were performed without overexpression when HeLa cells and MCF7 cells were used (Fig 2D and Fig 3B, 3C), suggesting protein-protein interactions at endogenous protein levels of SARAF and ALG-2. In the case of empty vector transfection in HEK293 cells, we also observed an interaction between endogenous SARAF and ALG-2 (Fig 2B, vector, Ctrl IgG vs anti-SARAF). Overexpression of SARAF increased the interaction with ALG-2 as expected. However, unfortunately as commented by Reviewer 2, we did not show interaction between ALG-2 and SARAF in cells on the spot by cell biological methods. To our regret, the obtained antibody against SARAF could not be used for immunofluorescence microscopic analysis. Alternatively, we performed fluorescence microscopic analysis to observe subcellular co-localization of fluorescent protein-fused proteins (SARAF-SGFP2 and mCherry-ALG-2). Please see a supplementary material for evaluation by Reviewer 2. We observed a partial co-localization of SARAF-SGFP2 and mCherry-ALG-2 after stimulation of HeLa cells with calcium-inducing reagent, thapsigargin. Unfortunately, the resolution was poor and this was the result of transiently overexpressed proteins. Establishment of stable cell lines expressing the fluorescently labeled proteins at appropriate levels would be necessary. We would like to perform more confident cell biological studies including FRET, if possible, for calcium-dependent SARAF-ALG-2 interaction and spatio-temporal association of ALG-2 with SARAF and the SOCE apparatus in cells in our future studies. This was added in Discussion in the revised manuscript.       

Round 2

Reviewer 2 Report

The revised paper is acceptable.